# Self-organization of human dorsal-ventral forebrain structures by light induced SHH

Riccardo De Santis[1], Fred Etoc[2], Edwin A. Rosado-Olivieri[1] & Ali H. Brivanlou [1✉]

Organizing centers secrete morphogens that specify the emergence of germ layers and the establishment of the body's axes during embryogenesis. While traditional experimental embryology tools have been instrumental in dissecting the molecular aspects of organizers in model systems, they are impractical in human in-vitro model systems to dissect the relationships between signaling and fate along embryonic coordinates. To systematically study human embryonic organizer centers, we devised a collection of optogenetic ePiggyBac vectors to express a photoactivatable Cre-loxP recombinase, that allows the systematic induction of organizer structures by shining blue-light on human embryonic stem cells (hESCs). We used a light stimulus to geometrically confine SHH expression in neuralizing hESCs. This led to the self-organization of mediolateral neural patterns. scRNA-seq analysis established that these structures represent the dorsal-ventral forebrain, at the end of the first month of development. Here, we show that morphogen light-stimulation is a scalable tool that induces self-organizing centers.

[1] Laboratory of Stem Cell Biology and Molecular Embryology, The Rockefeller University, New York, NY, USA. [2] Center for Studies in Physics and Biology, The Rockefeller University, New York, NY, USA. ✉email: brvnlou@rockefeller.edu

During embryogenesis, the central nervous system (CNS) is derived from an epithelial sheet of cells, the embryonic neural plate, that is induced in a polarized non-cell autonomous manner by a small group of cells called the Spemann organizer[1]. Neural induction activity of the organizer occurs by a default mechanism that is exerted through the secretion of soluble inhibitors that block both branches (SMAD1/5/8 and SMAD2/3) of the TGFb signaling pathway[2–6]. Dual SMADs inhibition directly converts pluripotent embryonic stem cells to the anterior neural tissue of the dorso-anterior forebrain[7]. Downstream of these primary neural inducing signals, highly localized and dynamic organizing centers provide multiple morphogen sources that pattern the CNS[8–10]. The embryonic neural tissue undergoes antero-posterior (A-P) and medio-lateral (M-L) patterning during neural plate stages[11,12]. As the neural tube closes, cells in the lateral part of the plate become dorsal (roof plate) and those in the midline become ventral (floor plate). Thus, M-L patterns are converted to dorso-ventral polarity (D-V).

The interplay between two signaling pathways, BMP4 and SHH, both acting as morphogens guide the establishment of the M-L and D-V polarity of the embryonic neural tissue. BMP4, is expressed in the lateral edge of the neural plate, and subsequently in the dorsal neural tube, while SHH, is expressed in the ventral midline of the neural plate, and subsequently in the floor plate of the neural tube[13]. Classical experimental embryology approaches such as ectopic presentation of SHH ligand by grafting coated beads, embryonic explants, or morphogen-secreting cells in mouse, chick or human stem cells demonstrated that SHH activity is sufficient to induce ventral neural fates via its transcriptional effector Gli3[14–18].

While these approaches have been instrumental in shaping our current understanding, they also suffer from technical shortcomings that have hindered a precise mapping of fate acquisition as a function of signaling dynamics in the context of early human development. For example, grafting experiments provide little control over the extent of the inductive field's spatial limits, control of throughput, and non-specific effects due to wound healing. The development of tools that systematically control these parameters will lead to better dissection of morphogens patterning and will constitute a major step forward in experimental embryology. Optogenetic tools have been recently used to control gene expression with spatiotemporal resolution, taking advantage of different strategies[19–25]. This carries the potential of creating exogenous embryonic organizer centers in model tissues for quantitative studying of embryonic induction and for creating in vitro self-organized structures that present the axial organization, that is a key landmark of embryonic development. Light modulation of signaling pathways provides flexibility and high spatial resolution over the morphogenetic stimulus[26–29].

Here, we have devised a collection of optogenetic ePiggyBac vectors to conditionally express a photoactivatable Cre-loxP recombinase for creating spatially restricted organizing centers that break symmetry in self-organizing hESCs. This collection is an hESCs optimized blue-light inducible split-CRE system based on the Magnets-split-CRE[20]. This system allows precise spatiotemporal control over the expression of a morphogen under a blue-light input. This experimental setup provides a highly quantitative and simplified method based on blue-light stimulation that can be used to induce organizing centers in in vitro cultures of hESCs. To establish proof of feasibility, we tested our tool for its ability to break symmetry in hESC-derived neural tissue to light-induce M-L polarity by inducing stripes of SHH expression as observed in the midline of the neural plate in embryo. Light induced and polarized expression of SHH during neural induction in absence of exogenous WNT modulation, led to the self-organization of a 2D in vitro human dorsal-ventral forebrain structure, that include a ventral telencephalic-hypothalamic primordia. Polarization of morphogens using light provides a non-invasive approach to decipher the earliest events that underly symmetry breaking in the embryonic nervous system in stages of human development otherwise inaccessible for scrutiny.

## Results

**Engineering a collection of optogenetic ePiggyBac vectors**. In order to provide light modulation of gene expression to human developmental studies, we re-engineered our original transposon ePiggyBac vector[30,31] to conditionally express a light-inducible Cre-recombinase enzyme that takes advantage of the Magnets dimerization system (Magnet-CRE)[20] (Fig. 1A). This allows for a stable integration of a blue-light dependent CRE enzyme in the genome. To avoid culturing cells in the dark, minimize leakage and to gain better control of light-sensitivity, we controlled the light-CRE enzyme using a Dox-inducible promoter and a second T2A peptide to improve the separation of its components (Fig. 1A, left panel). We paired this vector with a receiver ePiggyBac that carries two sequential ORFs (Red and Green modules) to be regulated by LoxP recombination in a mutually exclusive manner (Fig. 1A, right panel). Both vectors were stably transfected into our female XX hESC line, RUES2 (NIHhESC-09-0013) and stimulated with DOX and blue-light through a light-blocking photomask (Fig. 1B, Supplementary Fig. 1A–C). Dox- and Blue-light induced hESCs showed robust expression of the Green module in patterns imposed by the shape of the photomask (Fig. 1B, C, Supplementary Fig. 1B, C). The Green module expressing cells showed a highly reproducible pattern of expression that tightly correlate with the photomask that slightly increase over time, probably due to cell proliferation (Supplementary Fig. 1D). The efficiency of light conversion by single-cell fluorescence measurement shows that 24 h of blue-light stimulation converts 78.3% of the total cells, while controls kept in the dark, or in absence of Dox, show less than 1% of Green positive cells (Fig. 1D). The activation of the green module depends on the duration of blue-light stimulation. Pulsed blue-light for 600 cycles (24 h) is the most efficient treatment without inducing cell death, as measured by CASP3 activation (Fig. 1D, Supplementary Fig. 1E). The blue light-dependent induction of the Green module was also validated by measuring RNA levels by qRT-PCR in whole-illuminated samples (Fig. 1E). LoxP genomic recombination upon Dox and light treatment is shown by amplicon Sanger sequencing of the selected genomic region (Fig. 1F). The light-induced pattern of gene expression is consistently and stably maintained over six days in the absence of continuous light stimulation (Supplementary Fig. 1F). Light induced gene expression modulation was not confined to a single hESC line as another of our hESC line (RUES1, genetic background, male XY, NIHhESC-09-0012) responded in the same manner (Supplementary Fig. 1G). Collectively, these experiments demonstrate that pairing a light inducible CRE enzyme with a stable and drug inducible ePiggyBac vector allows for a rapid and efficient spatiotemporal control of exogenous gene expression in hESCs.

**Light induced dorso-ventral hESC-derived neural tissue**. In order to study the M-L and D-V aspects of neural patterning in human models, we tested the ability of light stimulation to generate a localized SHH organizing center. We devised a LoxP inducible Green module to be co-expressed with SHH at the mRNA level, while producing two separate proteins using a T2A peptide. This setup can be regulated by blue-light stimuli through LoxP recombination (Fig. 2A). hESCs (RUES2) were differentiated using dual SMADs inhibition (SB431542 and

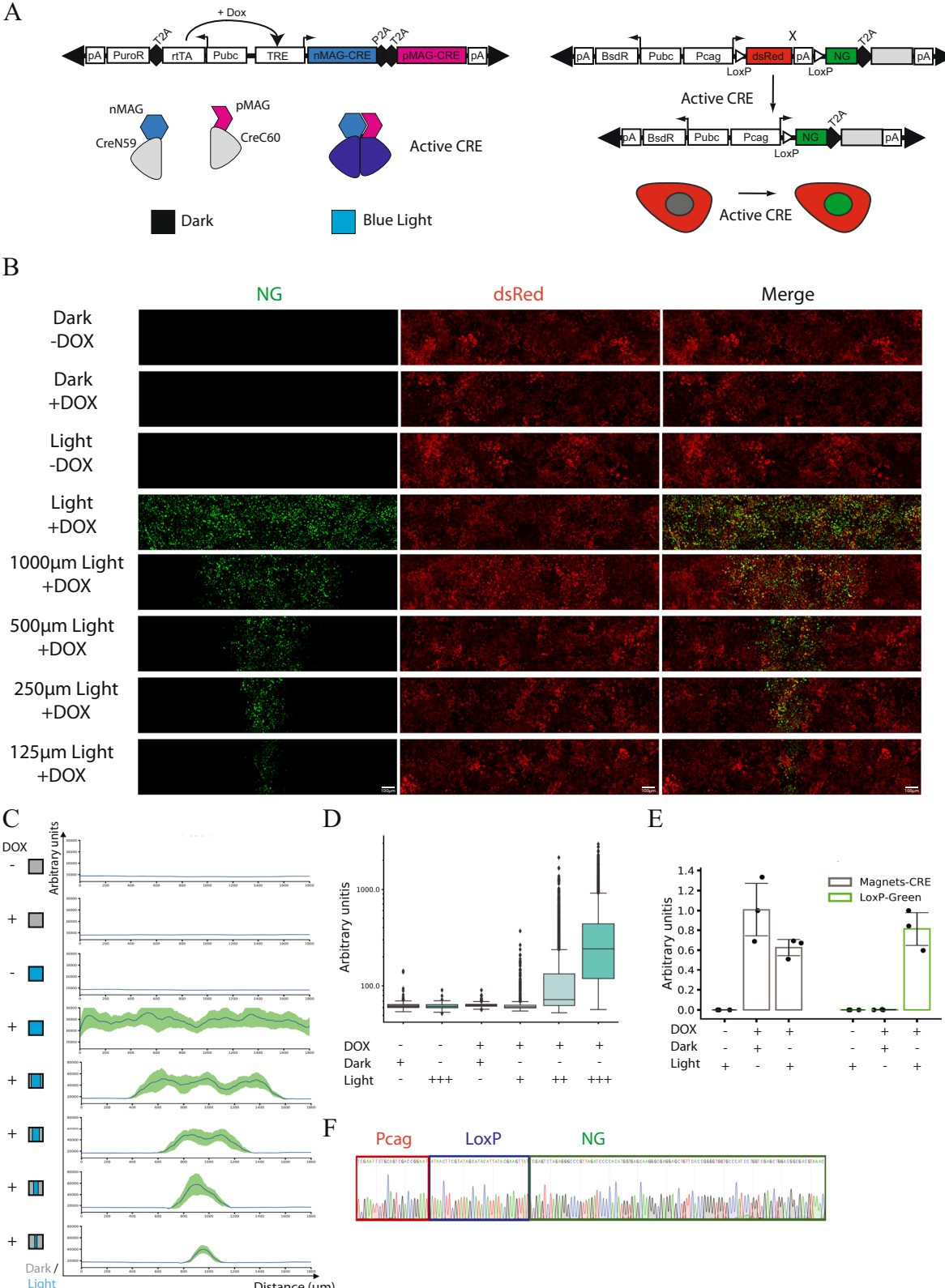

population, FOXA2 the floor plate and NKX2.1 the ventral neural progenitors in the basal telencephalon and hypothalamic primordia at E12.5 (Supplementary Fig. 2A)[16,32].

Blue light was shone on day 1 during neural induction through a 1 mm rectangular mask for 24 h to produce SHH and a green fluorescent protein (NG-T2A-SHH) or a control fluorescent protein (NG-CNTRL) in a spatially restricted domain. Light

LDN193189) to induce an anterior forebrain fate. The application of DOX at day 0 confers light sensitivity for the first two days of differentiation (Fig. 2A). Neural induction using dual SMADs inhibition has been shown to generate progenitors representative of the embryonic neuroepithelium[32–34]. At this stage, a set of defined markers can be used to decipher patterning at various distances from the SHH source: PAX6 marks the most dorsal

**Fig. 1 Light-induced gene expression programs in hESCs. A** Schematization of the collection of optogenetic ePiggyBac vectors to conditionally express a photoactivatable Cre-loxP recombinase vector. Left panel. Puromycin selectable and Dox inducible piggyBac transposon carrying the CRE-MAGNETs system. Dox treatment induces the CRE-MAGNETs system which reconstitutes a fully active CRE in the presence of blue light. Right panel. Blasticidin selectable LoxP exchangeable dual colors piggyBac transposon. The first ORF (Red module) is constitutive expresses while the second ORF (Green module) depends on CRE-LoxP recombination. Gray Box is the possible cargo protein taking advantage of the T2A peptide. **B** Photomask induced light patterns showing the expression of the Red module (dsRed), Green module (NG) and merge composite channel. Dark control and spatially localized activation in presence and absence of DOX with different spatial features are shown (1000 µm, 500 µm, 250 µm, 125 µm). Scale bar: 100 µm. **C** Cumulative fluorescence intensity analysis (line profile) over the $x$-axis. $x$-axis displays the linear distance in µm. $y$-axis reports the cumulative fluorescent intensity profile in arbitrary units. Line profile shows the average (line) and SD (area) for the Green fluorescent channel. Quantification after 600 cycles (24 h) of pulsed Blue light ($n = 3$ biologically independent samples). **D** Single-cell fluorescent intensity quantification. Conditions: presence or absence of Dox with concomitant Dark or Light stimulation. We titrated the time of light stimulation using different pulsed light conditions. 1 cycle of pulsed light is equal to 20 s Light-ON and 120 s Light-OFF. The light stimulation intervals are 1 cycle, 25 cycles (1 h), and 600 cycles (24 h), data are displayed as a boxplot where center lines show the medians, box limits indicate the 25th and 75th percentiles and whiskers extend to minimum and maximum values ($n = 5$ biologically independent samples). **E** qRT-PCR showing mRNA induction of the MAGNETs system and light-induced expression of the Green module, data are displayed as mean and SD ($n = 3$ biologically independent samples). **F** Sanger sequencing of the genomic region flanking the LoxP site showing the recombination of the Red module and the generation of the Green module (Red box: Pcag-promoter, Blue box: LoxP, Green box: NG).

stimulation induced the expression of the Green module in both NG-CNTRL and NG-T2A-SHH lines (Fig. 2B). At day 2, FOXA2 positive cells were specifically induced by the NG-T2A-SHH but not in the NG-CNTRL line (Fig. 2B, C, Supplementary Fig. 2B). FOXA2 positive cells were detected in cells secreting SHH as well as in cells just next to the SHH secreting domain, providing functional evidence of autocrine as well as paracrine SHH activity (Fig. 2B, C, Supplementary Fig. 2B, C). Examination of the NG-T2A-SHH light-induced cells after 7 and 14 days display FOXA2[+], NKX2-1[+] and PAX6[+] cells that are organized in discrete domains while NG-CNTRL cells acquire PAX6[+] default neural fate, in absence of ventral cell types (Fig. 2D–F; Supplementary Fig. 3A–C). The expression NG-T2A-SHH is stably maintained in absence of light over the course of the differentiation (Supplementary Fig. 3D, E). SHH light induction unveiled M-L self-organization of the neural populations under the influence of an organizing center. Interestingly, at day 7, a population of cells co-expressing FOXA-2 and NKX2-1 was detected within and near by the light-induced organizer (Fig. 2D–F, Supplementary Fig. 3A). These FOXA2[+]/NKX2-1[+] double-positive cells have been suggested to be human specific, as have not been detected in the mouse, while they are present in the ventral forebrain in human fetal samples at PCW5.5[16]. PAX6, NKX2-1 and FOXA2 domains gradually segregate over time inside and outside the SHH induced domain, with PAX6[+] cells localized the farthest from the SHH source (Fig. 2D–F). Single-cell quantification shows that at day 7, the organizer induces population of cells double positive for NKX2-1[+]/FOXA2[+], both cell autonomously and non-cell autonomously (Fig. 2E, upper panel). A fraction of light converted cells, express high levels of FOXA2 but not NKX2-1 (Fig. 2D, E, upper panel). At day 14, the ventral cellular populations induced by SHH differentiate into a NKX2-1[+]/FOXA2[−] population, that is located both laterally and inside the light induced SHH organizer (Fig. 2D, E, lower panel). Also, there is a population of cells NG-T2A-SHH[+]/FOXA2[+]/NKX2-1[−] (Fig. 2E, F, Supplementary Fig. 4A). NKX2-1 domain juxtaposed to the SHH organizer is induced independently from the size of the SHH domain (Supplementary Fig. 4B). The RNA expression of the NG, the exogenous SHH and its downstream target GLI1 correlate with the expression of the NG-T2A-SHH module, validating our co-expression strategy (Supplementary Fig. 4C). Therefore, our analysis revealed a proximal distal pattern of ventral cell fates from the SHH organizer during neural induction. Spatiotemporal control of SHH induces ventral neural fates that are organized in a 2D space in vitro, resembling M-L and D-V neural populations (Supplementary Fig. 2D). Moreover, it validates the functionality of our optogenetic tool for its ability

to induce and self-organize discrete fates in hESCs with a simple blue light stimulation.

**Molecular characterization of light-induced neural fates.** In order to precisely and unbiasedly identify the cell types present in our light-induced, self-organizing neural tissue, we characterized their transcriptome using scRNA-seq from two independent biological replica (12097 and 6207 cells) (Fig. 3A, B, Supplementary Fig. 5A–C). Cells were differentiated and stimulated with blue light as previously described in Fig. 2A and harvested at day 14 for single-cell RNA-sequencing analysis. Differentially expressed genes based on leiden clustering, RNA-velocity trajectories and cell-cycle predictions were used to classify 14 distinct cell identities: (i) FOXA2[+]/ARX Floor Plate; (ii) NKX2-1[+]/RAX[+]/SIX6[+] Ventral tuberal hypothalamic progenitors; (iii) NKX2-1[+]/FOXG1[+] Ventral telencephalic progenitors, (iv) NKX2-1[+]/NHLH2[+]/OTP[+] Ventral hypothalamic neurons; (v) TFAP2A[+]/KRT19[+] Superficial ectoderm; (vi) SOX10[+]/PLP1[+] Neural Crest, (vii) PAX6[+]/EMX2[+]/OTX2[+] Dorsal forebrain progenitors; (viii) IRX3[+]/OLIG3[+] Dorsal thalamic progenitors; (ix) TBR1[+]/LHX1[+] Dorsal Neurons_1; (x) HES6[+]/DLL3[+] Dorsal Neurons_2; dorsal and ventral proliferating progenitors, (xi) Dorsal, (xii) Dorsal thalamic, (xiii) Ventral and (xiv) an unidentified population (UnId) (Fig. 3B, Supplementary Fig 5C-D, Source Data file). This analysis revealed the presence of multiple cell types that demarcate different domains along the embryonic D-V and A-P axes in agreement with what was previously shown by asymmetric SHH stimulation in 3D organoids[18]. No endoderm, mesoderm or extra-embryonic markers were detected, locating cells in the ectodermal compartment.

To correlate the timing of our self-organizing in vitro tissues with in-vivo events, we integrated our dataset with a scRNA-seq collection of mouse brain samples at different time points, E8.5, E10, E12, E12.5, E13, and E15[35]. scRNA-seq transcriptomics of the human light-induced cells, grouped as neural precours (NPCs), floor plate, superficial ectoderm, neurons and UnId cells, integrate with the in-vivo mouse brain developmental atlas (Supplementary Fig. 6A). The human NPCs display high correlation with the Radial glia population at E10, while the in vitro derived neurons display high correlation with the mouse neuronal category at E12.5-E15 (Supplementary Fig. 6B, C), suggesting a temporal match with human development at PCW4-5 (https://embryology.med.unsw.edu.au/embryology/index.php/Carnegie_Stage_Comparison).

The expression of GLI3, GAS1 and PTCH was used to identify SHH receiving cells. In agreement with literature, we identified a

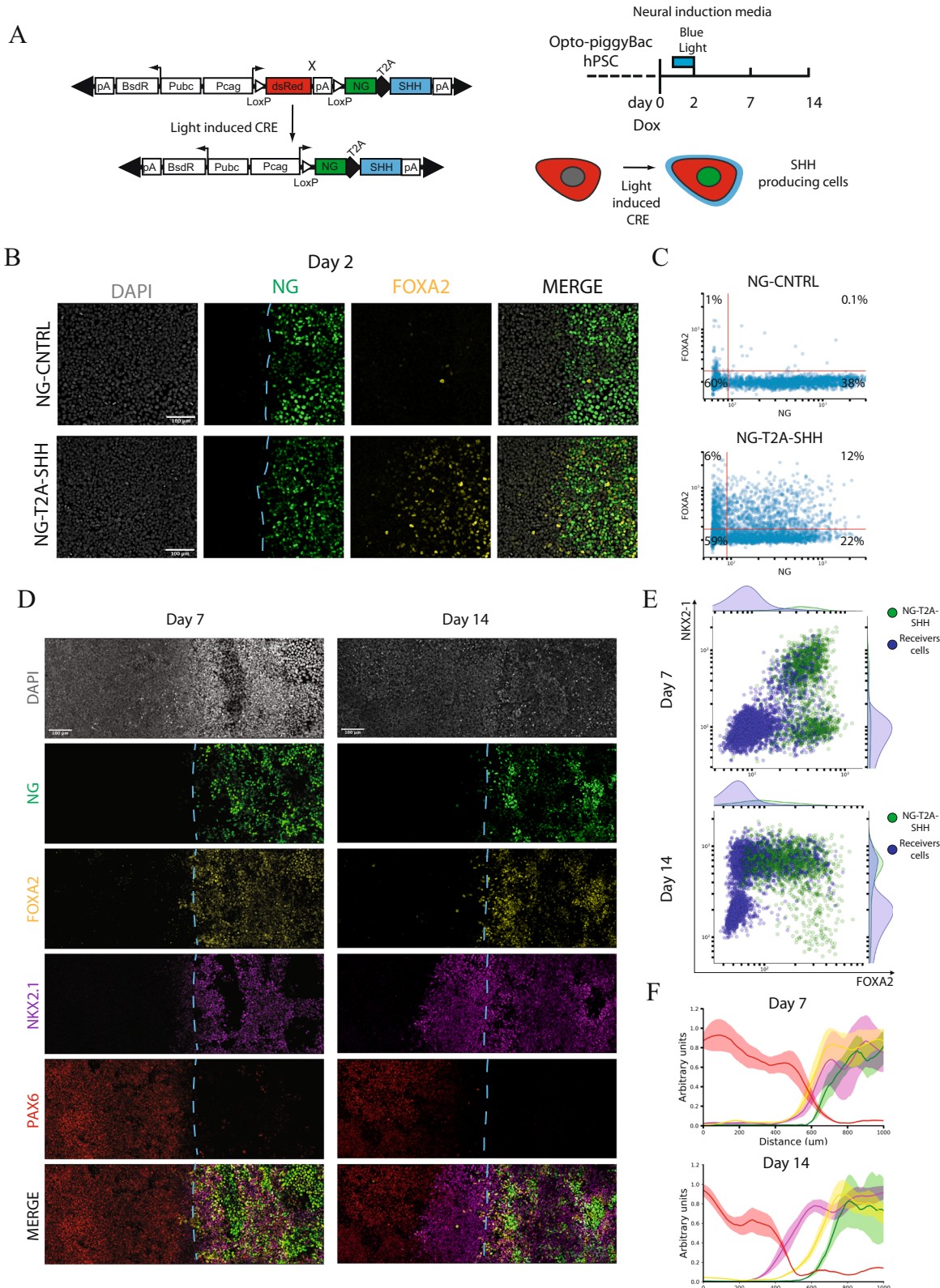

GLI3-GAS1^low/PTCH1^high population as SHH stimulated while GLI3-GAS1^high/PTCH^low cells as unstimulated (Fig. 3C, Supplementary Fig. 7A). We also validated the specific induction of SHH signaling in the NG-T2A-SHH line compared to a NG-CNTRL line by testing GLI1 and GLI3 expression at day 14 using qRT-PCR (Fig. 3D). Among the SHH induced populations, scRNA-seq analysis confirmed the presence of a FOXA2⁺/ARX⁺, floor plate

population and revealed the identity of four distinct NKX2-1⁺ populations (Fig. 3B, C, E). The first is representative of the tuberal hypothalamic neural progenitors positive for NKX2-1, SIX6, SIX3, RAX (Fig. 3C, Fig. 4A–C, Supplementary Fig. 7C)[36–38], the second is the NKX2-1⁺ FOXG1⁺ population representative of the ventral telencephalic population (Fig. 4A, B)[16,18], the third is a ventral population of proliferating

**Fig. 2 Medio-lateral neural patterning by light-induced SHH. A** Left panel. Schematization of Green module T2A human SHH. This setup allows conditional expression of SHH and fluorescent visualization of the light-induced cells. Right panel. Schematization of Blue-light stimulation and neural induction reporting analyzed time points. Neural induction is induced by inhibiting TGFβ signaling by dual SMADs inhibition that is maintained during the entire course of differentiation (14 days). Dox treatment starts at day "0" and it is washed-out at day "2". **B** FOXA2 immunostaining in light induced SHH cells and NG-CNTRL during differentiation at day 2. DAPI-Gray, NG-Green, FOXA2-Yellow, Merge-Composite. Dashed cyan line indicates the border of SHH producing cells. Scale bar: 100μm. **C** Day 2 single-cell fluorescence quantification displayed as a scatterplot, reporting FOXA2 and NeonGreen (NG) intensity for NG-CNTRL and NG-T2A-SHH. **D** Immunostaining time-course analysis of the dorsal and ventral fates, revealing NG-Green (Organizer), FOXA2-Yellow (Ventral floor plate), NKX2-1-Magenta (Ventral neural progenitors), PAX6-Red (Dorsal neural progenitors), Merge-Composite at day 7 and day 14 during neural differentiation in response to ligh patterned SHH. Dashed cyan line indicates the border of SHH producing cells. Scale bar: 100 μm. **E** Single-cell fluorescence quantification of NG-T2A-SHH induced cells displayed as combined scatterplot and density histogram at day 7 (top) and day 14 (bottom). *x*-axis report FOXA2 while the y-axis the NKX2-1 fluorescence intensity profile. Each dot represents an individual cell that has been color coded according to its green module status. The green dots represent the NG-T2A-SHH positive cells, while the blue dots are NG-T2A-SHH negative cells. **F** Cumulative fluorescence intensity analysis (line profile) over the *x*-axis. *x*-axis displays the linear distance in μm. *y*-axis reports the cumulative fluorescent intensity profile in arbitrary units for each channel. Line profile shows the average (line) and SD (area) for each channel. The line profile is color-coded as the immunofluorescent channels, NG-Green, FOXA2-Yellow, NKX2-1-Magenta, PAX6-Red. Top panel. Day 7 quantification (*n* = 4 biologically independent samples). Bottom panel. Day 14 quantification (*n* = 4 biologically independent samples).

progenitors and the fourth is a small population of ventral neurons that we classified as hypothalamic neurons positive for NKX2-1, OTP, NHLH2 (Fig. 3B, C, Source Data file)[37,38]. Among the SHH unstimulated cells, we identified dorsal populations that consist of forebrain progenitors (PAX6$^+$/EMX2$^+$/OTX2$^+$), thalamic progenitors (IRX3$^+$/OLIG3$^+$), two neuronal populations (TBR1$^+$/LHX1$^+$ and HES6$^+$/DLL3$^+$), and non-neural ectoderm derivatives such as superficial ectoderm and neural crest (Fig. 3B, Supplementary Fig. 5C, Source Data file). The non-neural ectoderm population derive mostly from the plate edge independently from the light organizer (Supplementary Fig. 7B).

Immunostaining for specific markers, SIX6, RAX, NKX2-2 and FOXG1, revealed the spatial segregation of telencephalic and hypothalamic territory (Fig. 4B, C, Supplementary Fig. 7C, D, E). We further showed that light-modulation of SHH not only self-organizes telencephalic and hypothalamic progenitors, but also neurons, since hypothalamic OTP$^+$ neurons are preferentially located proximal to the light-induced organizer (Fig. 4D). While the differentiation of hypothalamic cells has been previously observed in traditional cell culture or 3D organoids[39–41], confinement of a SHH source in 2D instructs the self-organization of a ventral telencephalic-hypothalamic structures that are spatially organized in monolayer.

Finally, the hypothalamic marker genes used in this study were in-vivo validated for their specific expression in the human fetal hypothalamus at PCW10 (Supplementary Fig. 8A)[42]. In order to capture the gene expression modules that are shared between our in vitro dataset and the fetal hypothalamus, we computed the gene regulatory networks (regulons) in each dataset using pySCENIC[43]. Among the 427 active regulons identified in the human fetal hypothalamus dataset, the 72.5% (310 regulons) are shared with our in vitro dataset (Supplementary Fig. 8B, Source Data file). Based on RNA velocity analysis, we identified in the light-induced scRNA-seq dataset a ventral differentiation trajectory that starts from the ventral proliferating progenitors and ends at the ventral hypothalamic neurons (Fig. 3B). Performing gene ontology analysis of genes that are differentially expressed along this trajectory, we identified waves of gene expression linked to cell-cycle regulation, neural progenitor expansion and neuronal maturation (Supplementary Fig. 8C). We explored whether some important gene expression patterns recently described in the context of human fetal hypothalamic development were recapitulated in our model, TTYH1, HMGA2 and MYBL2 show the same progenitor-neuronal trend observed in the human fetal hypothalamus at PCW10 (Supplementary Fig. 8D).

Collectively, our experiments demonstrate the ability to generate organizing centers by a simple blue light-induction of a morphogen, which specify a morphogenic source that patterns discrete cell types along a proximal distal axis in space, and lineage trajectory in time.

## Discussion

We devised a collection of optogenetic ePiggyBac vectors to conditionally express a photoactivatable Cre-loxP recombinase, that allows the control of morphogen expression using a simple blue light stimulation in hESCs. In this study, we used this tool to induce a localized domain of SHH expression to generate a ventral organizer within a neutralizing tissue. This establishes proof of feasibility for a technological concept, which we predict to apply to a variety of gene expression modules, including signaling pathways, transcription factors and cell-cycle players. This experimental embryology tool has advantages over the classical techniques as it provides higher throughput and resolution. This system allowed us to control the asymmetric induction of the morphogen SHH during hESCs differentiation, which generated multiple locally organized cell types. It represents the first generation of light-induced self-organization of cell fates and patterns in hESCs. Future optimization will improve several aspects. For example, it is foreseeable to add a pulsatile temporal regulation to this setup by imposing another layer of regulation rather than a step induction as it is in this study. Traditional methods for inducing local perturbation of signaling pathways, such as beads, provide little control over the extent of the inductive fields and are incompatible with high-throughput setups. Our in vitro model system displays high reproducibility and spatial control among many independent wells of differentiating human pluripotent stem cells. Since our light-inducible experimental setup is unidirectionally activated, the forebrain D-V structures obtained in this study, are likely generated by the combination of direct SHH signaling and proliferation/expansion of the induced progenitors. We envision that manipulating multiple signals using different light wavelength provides the promise of more sophisticated inductive interaction studies[44,45]. For example, pairing ligands and inhibitors to precisely decipher the cellular and molecular aspects of Turing based reaction diffusion events, which have been shown to have an instructing role in the establishment of embryonic patterns.

Light control of gene expression allows geometric confinement of the stimulus and its response in inducing the ventral organizer SHH. We have previously shown that physical geometrical confinement of hESCs is sufficient to induce self-organization in response to BMP4 in the context germ layers (gastruloids)[46]. Self-organization has also been demonstrated to occur in the context of a single embryonic germ layer, ectoderm[34,47]. Dual SMADs

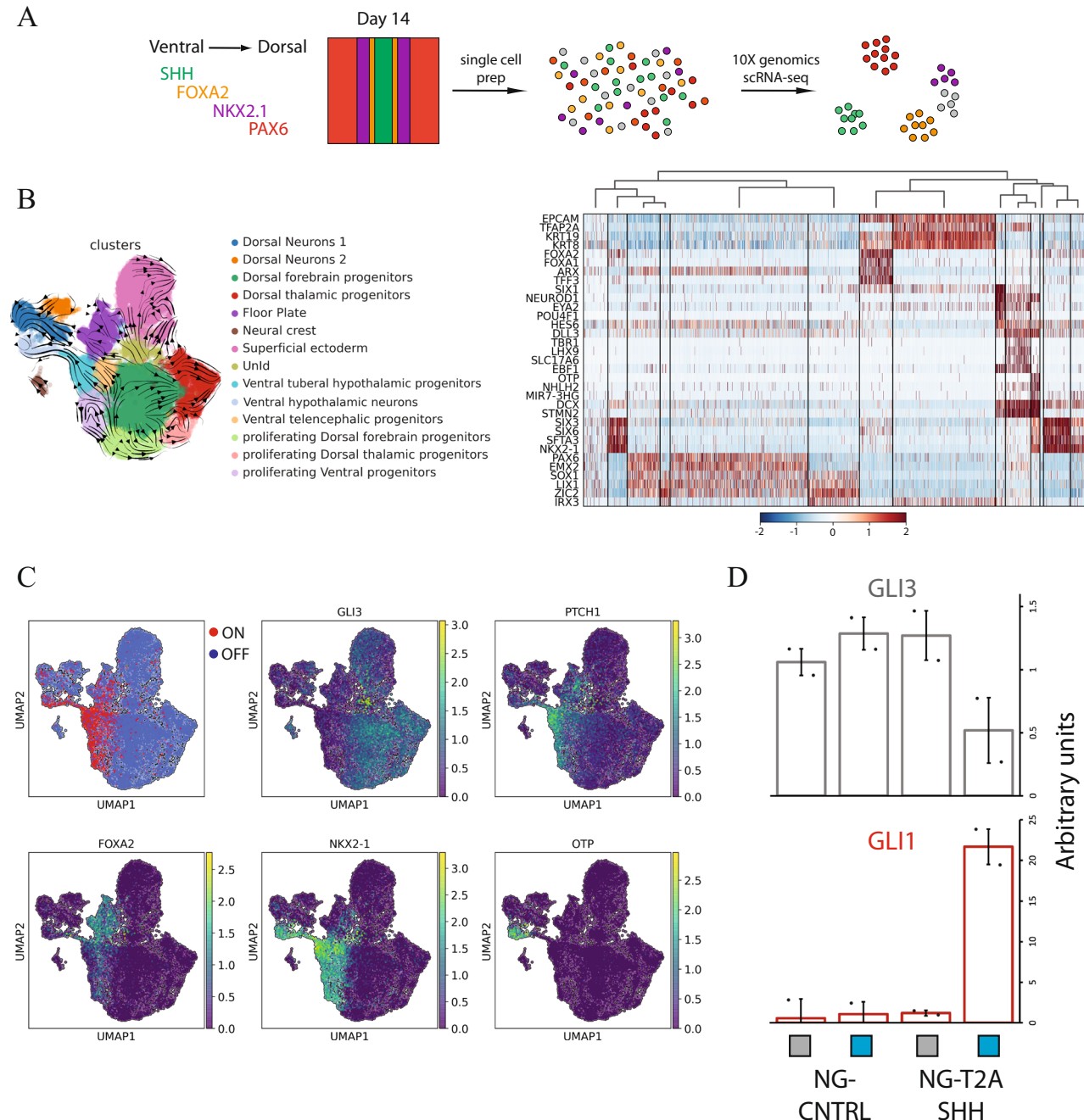

**Fig. 3 scRNA-seq characterization of light-induced SHH neural cells. A** Schematization of the scRNA-seq profiling strategy. **B** Left panel: UMAP plot labeled with the identified cell identities and RNA velocity vectors: (i) FOXA2+/ARX Floor Plate, (ii) NKX2-1+/RAX+/SIX6+ Ventral tuberal hypothalamic progenitors, (iii) NKX2-1+/FOXG1+ Ventral telencephalic progenitors, (iv) NKX2-1+/NHLH2+/OTP+ Ventral hypothalamic neurons, (v) TFAP2A +/KRT19+ Superficial ectoderm, (vi) SOX10+/PLP1+ Neural Crest, (vii) PAX6+/EMX2+/OTX2+ Dorsal forebrain progenitors, (viii) IRX3+/OLIG3+ Dorsal thalamic progenitors, (ix) TBR1+/LHX1+ Dorsal Neurons_1, (x) HES6+/DLL3+ Dorsal Neurons_2, dorsal and ventral proliferating progenitors, (xi) Dorsal, (xii) Dorsal thalamic, (xiii) Ventral and (xiv) an UnIdentified population (UnId). Right panel: z-score scaled heatmap of marker genes used for cell identities classification. **C** UMAP plot displaying SHH responsive and unresponsive status, and GLI3 and PTCH1 expression level. SHH responsive and unresponsive status have been computed imposing a threshold based on the normalized distribution of GLI3, GAS1 and PTCH1 counts, where GLI3-GAS1low and PTCH1high represent SHH responsive while GLI3-GAS1high, PTCH1low unresponsive. **D** qRT-PCR analysis of SHH responsive genes (GLI1 and GLI3) for NG-T2A-SHH and NG-CNTRL at day 14 of differentiation. The histogram displays the average and SD of differentiated cells exposed to light stimulation or dark control ($n = 2$ biologically independent samples). .

inhibition leads to self-organization of telencephalic neural progenitor around a lumen generating rosette (cerebroids)[34,47]. When Dual SMADs inhibition is followed by BMP4 presentation, confined hESCs colonies self-organize to generate the four derivatives of the ectodermal layers: neural, neural crest, sensory placode and epidermis that organize in radially symmetrical patterns[34]. In the context of the ectoderm, this type of self-organization reflects the events that occur in the dorsal anterior part of the developing CNS. As for spatial geometrical confinement that provided the key element for gastruloid, cerebroid and

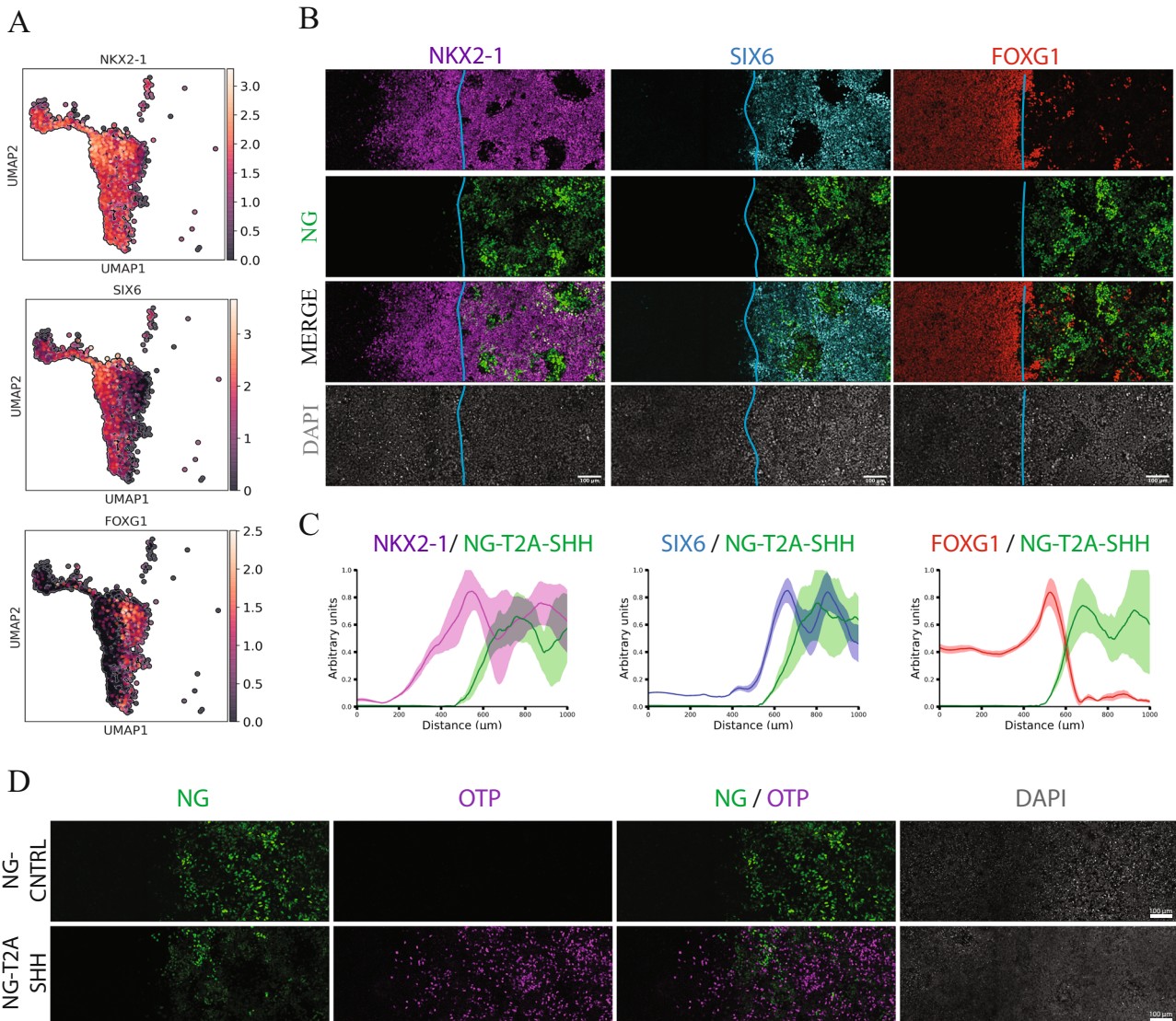

**Fig. 4 Spatial self-organization of telencephalic and hypothalamic fates upon light-induced SHH. A** UMAP plot showing a selected population of cell NKX2-1+. Expression of markers that distinguish hypothalamic and telencephalic populations (NKX2-1, SIX6 and FOXG1). **B** Immunostaining shows the spatial segregation of ventral population arising from a light-induced SHH source at day 14. (Green NG, Magenta NKX2.1, Cyan SIX6, Red FOXG1, Gray DAPI), Scale bar = 100 μm. **C** Cumulative fluorescence intensity analysis (line profile) over the x-axis. x-axis displays the linear distance in μm. y-axis shows the cumulative fluorescent intensity profile in arbitrary units for each channel. Line profile shows the average (line) and SD (area) for each channel. The line profile is color-coded as the immunofluorescent channels, NG-Green, NKX2-1-Magenta, SIX6-Cyan, FOXG1-Red at day 14 quantification ($n = 3$ biologically independent samples). **D** Immunostaining shows OTP positive cells induced in proximity of the NG-T2A-SHH organizer but not in the NG-CNTRL. Light induced SHH drive the self-organization of both neural progenitors and neurons (Green NG, Magenta OTP, Gray DAPI), Scale bar = 100 μm.

neuruloids self-organization, here we confined the geometry of a morphogenic organizer by imposing a chemical edge with light stimulation. Complementing these results, our tool allowed the self-organization of the ventral aspect of the developing nervous system. It is tempting to speculate that self-organization imposed by geometrical confinement when combined with spatial manipulation of a morphogen will lead to more sophisticated and complex aspects of self-organization. Confinement of geometry and organizer structures promise to generate highly quantitative models that will more faithfully represent early human development in vitro.

We and others have shown that it is possible to deconvolve several aspects of embryonic self-organization by modeling developmental events in vitro. Our study uses a light inducible tool to generate 2D models of D-V forebrain development, composed of neural telencephalic and hypothalamic populations.

Interestingly, the hypothalamus is among the most conserved structures of the brain in vertebrates. In the adult, it is composed of several spatially distinct nuclei that control a wide range of functions, from body homeostasis to behavior and it is connected with the endocrine system through the pituitary gland. To the best of our knowledge, a comprehensive and systematic induction of discrete hypothalamic nuclei in the human context has not been achieved yet. Spatiotemporal induction of specific signaling pathways in the context of hypothalamic self-organization promise to recapitulate, in vitro, several aspects of the highly spatially organized hypothalamic development. Self-organization follows self-organization, beginning from an entire embryo (gastruloid) to a single embryonic germ layer such as the ectoderm (cerebroids and neuruloids), and ultimately the formation of discrete organs and cell types (dorsal and ventral brain). By this logic we expect that the development of discrete hypothalamic nuclei will also

follow the rules of self-organization and symmetry breaking. We speculate that the generation and patterning of hypothalamic territories are possible as long as the appropriate physical (confinement) and chemical (morphogen) cues are provided. Self-organizing models of the human hypothalamus in vitro will shed light on the basic principles that govern the development of a fundamental brain controlling center.

## Methods

**Human pluripotent stem cell culture**. hESCs lines used in this study are part of the NIH Human Embryonic Stem Cell Registry (RUES1-NIHhESC-09-0012) and (RUES2-NIHhESC-09-0013). Human pluripotent cells were maintained in feeder-free conditions on Geltrex-coated dishes. Cells were fed with MEF conditioned medium (CM) supplemented with bFGF 20 ng/mL daily[48]. Cells were routinely passaged every 3–4 days using Gentle Cell Dissociation Reagent (STEMCELL-Technologies).

**Neural induction for pluripotent stem cell**. hESCs were passaged as single cells using Accutase (STEMCELL-Technologies) and seeded in CM media supplemented with bFGF and Y27632 at 100,000 cells/cm$^2$ density. The differentiation started 1–2 days later when cells reached 100% confluency. Neural induction media is composed of: 50% Neurobasal and 50% DMEM-F12, supplemented with N2 0.5%, B27 0.5%, Glutamax 0.5X and Non-essential amino acid 0.5X and Insulin (2.5 μg/ml) all from LifeTechnologies. Small molecule SB431542 (10 μM) and LDN193189 (100 nM) were supplemented to warm media.

DOX (1 μg/ml) induction is used at day "0" concomitantly with dual-SMAD inhibition, and then washed out at day "2". Neural induction media supplied with dual-SMAD inhibitors is maintained throughout differentiation until day "14".

**piggyBac vectors generation**. One of the two piggyBac DOX inducible and Puromycin selectable vector was digested with HindIII-NotI and ligated with the Magnets-CRE expression cassette (pcDNA-MagnetsCRE plasmid is a kind gift from Dr. Yazawa). The other that is Blasticidine selectable, and carries a constitutive promoter (CAG), was modified to express a LoxP regulated switch of the two ORFs. A first "Red module" that express a dsRed protein, is removed and upon LoxP recombination allows the expression of a second "Green module", which express a Neon-Green protein fused with a T2A, allowing co-expression with other proteins (Fig. 1B). The LoxP-dsRed-LoxP was cloned using BamHI-BglII restriction site from the pLV-CMV-LoxP-DsRed-LoxP-eGFP, a gift from Jacco van Rheenen (Addgene plasmid # 65726). PCR-amplified SV40 poly-adenylation site was introduced using NheI restriction enzyme after the dsRed ORF. The Green module consisting of the nuclear localized Neon-green protein was synthesized using IDT geneblocks and inserted using the restriction enzymes AgeI-NotI. The SHH coding sequence was PCR amplified and cloned using BsmBI-NotI restriction enzymes from pOEM1 pCMV:hShh-pPH:vsvged, a gift from Elly Tanaka (Addgene plasmid # 111156). Plasmids used in this study have been deposited on ADDGENE.

**Stable integration of piggyBac vectors in the genome of hESCs**. hESCs were nucleofected using an Amaxa Nucleofector II (Lonza) according to manufacture's instruction for hESC nucleofection. A mixture (1 μg + 1 μg + 0.5 μg) of: ePB-PURO-TT-PA-CRE, ePB-BSD-CAG-RLOXP-NG-T2A or ePB-BSD-CAG-RLOXP-NG-T2A-SHH, and the piggyBac transposase were nucleofected in 10$^6$ hESCs and plated in Geltrex (Life technologies) coated plates using CM media supplemented FGF2 (20 ng/mL) and ROCK-inhibitor (Y-27632, 10 μM) for 24 h. Cells were then selected using CM media supplemented with blasticidin 5 μg/ml and puromycin 1ug/ml (all from Life technologies) for 2 weeks.

**Immunostaining**. Cells were washed in PBS−/− and fixed in PFA 4% for 30 min at room temperature (RT). After fixation, 96 or 24 wells plates were washed three times with PBS−/−. Blocking was performed in PBS−/−, TritonX (0.15%), Normal Donkey Serum 3% (Ab-solution) for 30 min at room temperature. Primary and secondary antibody staining was done at room temperature for 1 h each, in Ab-solution, followed by three washes in PBS−/− TritonX (0.15%). The list of primary antibodies is provided below. DAPI (Thermo Fisher Scientific) staining was performed 5 min and plates were stored in IBIDI mounting medium at 4 °C before imaging.

**Blue light-stimulation**. Cells were plated into IBIDI 24 or 96 wells plates and placed on top of a custom-made black plastic box. The bottom on the IBIDI plate was aligned with a custom-made laser cut plastic layer or with a film photomask that spatially defines the area of illumination. The Blue light sources used in this study are a custom LED-blue light tablet or the LED 96 well array system (AMUZA INC). Pulsed illumination was delivered to the samples with the following parameters: 20 s ON-time, 120 s OFF-time cycling for 24 h at 10 V power.

**scRNA-seq preparation**. scRNA-seq sample preparation was performed following manufacturer's instructions. In brief, cells were enzymatically dissociated as single cells using Accutase for 8 min at 37 °C, washed twice in PBS−/− BSA 0,04% and then strained with a 40 μm filter. Gem formation and library preparation was performed following manufacturer's instructions using Single Cell 3′ v.3 reagents (10X Genomics).

**scRNA-seq analysis**. The scRNA-library for light induced SHH at day 14 during neural induction was sequenced using a NovaSeq 6000 SP flowcell as 28 × 94 × 8. FASTQ files were aligned using Cell Ranger (v.2.0.2) against hg19 reference genome. Count matrix were further processed in Python using Scanpy environment[49] (https://pypi.org/project/scanpy/) and are available under the NCBI GEO accession number GSE163505. The raw matrices were filtered to have a minimum of 500 detected genes per cell and genes were filtered to be expressed in at least 5 cells. Cells with over 15% mitochondrial UMIs were discarded. Clustering analysis was performed using Leiden algorithm on normalized counts and visualized in a low-dimensional space using UMAP plots (n.neighbors = 50, PCA components=20). Marker genes used to annotate the clusters were identified using a Wilcoxon rank-sum (Mann-Whitney-U). The integration of scRNA-seq biological replicas has been performed by matching mutual nearest neighbors using the Scanpy external API MNN correct, with default parameters (svd_dim=50)[50].

**RNA velocity and pseudotime analysis**. Kallisto Bustools[51] was used to align and quantify layered spliced and unspliced gene counts. The two scRNA-seq replicas were aligned independently, using kb-python=0.24.4 to capture spliced and unspliced transcripts, applying the -lamanno function with the GRCm38 human genome. The Kallisto unfiltered count matrix was imported in Scanpy and further processed using scVelo (v0.2.2)[52], using UMAP coordinates and clusters annotation from previously annotated Scanpy object. Count matrixes carrying the spliced/unspliced layer annotation were filtered and normalized for minimum shared counts=30 and the top 2000 high variable genes. The first 30 principal components and 30 neighbors were used to calculate 'Ms' and 'Mu' moments of spliced/unsliced abundance. We used the steady-state model[53] to determine velocities using the "scv.tl.velocity" function with default parameters.

We computed pseudotime analysis on a subset of cell classified as NKX2-1$^+$ hypothalamic cells, consisting of: NKX2-1$^+$/RAX$^+$/SIX6$^+$ Ventral tuberal hypothalamic progenitors, NKX2-1$^+$/NHLH2$^+$/OTP$^+$ Ventral hypothalamic neurons and NKX2-1$^+$/TOP2A$^+$ ventral proliferating progenitors. Using the python wrapper of Slingshot, "scprep.run.Slingshot" (scprep 1.1.0, https://pypi.org/project/scprep/), we calculated the pseudotime order of cells using full covariance matrix and default parameters. The normalized and log transformed count matrix for gene trends investigation has been denoised using MAGIC (3.0.0)[54] (default parameters, $t = 3$). Trajectory scatterplots are displayed as a function of the pseudotime, $x$-axis, and the gene expression values, $y$-axis.

Gene patterns have been identified using the scaled high variable genes and the Agglomerative Clustering algorithm (from scikit-learn (0.22), https://scikit-learn.org/stable/) along the pseudotime (clusters Number=20) using default parameters. Selected gene pattern clusters have been tested for enrichment of specific categories using GSEAPY (0.10.5)-enrichr (GO biological process 2018) (https://pypi.org/project/gseapy/)[55].

**Analysis and integration of publicly available scRNA-seq datasets**. scRNA-seq samples from the mouse brain developmental atlas[35] were downloaded as a Loom file and sliced according to their column annotation (E8.5, E10, E12, E13, E15). Extracted count matrixes were further normalized, log transformed and filtered for mitochondria content less than 15% and genes to be expressed in at least 5 cells. Gene annotations were converted to their human orthologous using "scanpy.queries.biomart_annotations". UMAP coordinates were calculated using (n.neighbors = 50, PCA components= 30). Human and mouse datasets were integrated using Ingest "sc.tl.ingest" function, using the common expressed high variable genes between the two datasets, using default settings. Correlation analysis of the integrated dataset was performed using the "Class" annotation from the mouse annotation and the human light-induced cells, grouped as neural precursors (NPCs), floor plate, superficial ectoderm, neurons and UnId cells. Correlation analysis is calculated using the scanpy correlation matrix function with default pearson parameters. Dendrogram is calculated using default settings and 30 PCA.

scRNA-seq count matrix of the human fetal hypothalamus[42] was preprocessed using the following settings: genes expressed in more than 5 cells and cells with less than 20% of mitochondrial genes were used for further analysis. Count matrixes have been normalized and log transformed. UMAP coordinates and clusters were obtained using (n.neighbors = 10, PCA components= 30) and leiden clustering (resolution=1).

**Regulons identification**. pyScenic (0.11.2) is used to identify gene regulatory networks in scRNA-seq datasets, using default parameters[43]. In brief, count matrixes were preprocessed using the following parameters: minimum number of genes per cell = 200 and genes expressed in at least 3 cells. The filtered count matrix is used to infer co-expression modules using the human transcription factors ranked database (https://resources.aertslab.org/cistarget/databases/). The

Arboreto algoritm (grnboost2) is used to calculate adjacence matrixes. Regulons are computed using the adjacencies matrixes of each dataset independently, followed by the prune module for targets with cis regulatory footprints (RcisTarget). Commonly detected regulons between the fetal hypothalamic dataset and our scRNA-seq dataset have been tested for enrichment of gene ontology, using the "Tissues protein expression from the human proteome map" from GSEApy-enrichr.

**Imaging**. Confocal images were acquired on a Zeiss Inverted LSM 780 laser scanning confocal microscope with a ×20 dry objective.

**Imaging analysis**. Images were preprocessed to display the Maximum Intensity Projection (MIP) of at least 4 z-stacks using ZEN black software. Fiji_V2 (Version 2.1.0/1.53c) was used for image display and formatting. MIP images were subsequently imported into Python using the Czifile package (2019.7.2). Line profile analysis consists of a single channel exported and the intensity values for each pixel over the y-axis were summed and displayed as a curve. Relative internal normalization is performed when indicated. Line profile in a 0–1 range was obtained normalizing values for each channel according to their maximum and minimum value.

Single-cell fluorescence intensity quantification was performed by identification of individual cells using an Otsu binarized DAPI image followed by distance transform and Watershed algorithm (scikit-image, 0.16.2) to separate overlapping nuclei. The pixel intensity of each object is plotted.

Cell death is quantified by calculating the area of CASP3 positive staining normalized on the total area. CASP3 positive signal is obtained upon Otsu thresholding of CASP3 staining using the python package (scikit-image 0.16.2, threshold_otsu). Plotting was performed after data processing using numpy (1.19.5) and pandas (1.1.5) libraries, using matplotlib (3.2.2) and seaborn (0.11.2) plotting libraries.

**qRT-PCR**. RNA from individual wells was extracted using the Qiagen RNeasy Plus Mini Kit. 1ug of RNA was retrotranscribed using Transcriptor First Strand cDNA Synthesis Kit (Roche 04897030001) and Real-time quantification was performed using SYBR Green Master Mix (Roche 04887352001). HPRT or ATP5O endogenous control is used for internal normalization and results are expressed as fold change over a reference sample.

**Sanger sequencing**. PCR-amplified amplicons (Hotstart Q5 polymerase, NEB) were sanger sequenced using Genewiz sanger sequence service under standard manufacture instructions.

**Images from the Allen brain atlas**. Images were adapted from the mouse Allen brain atlas showing ISH staining. Image credit: Allen Institute (https://developingmouse.brain-map.org)[56].

The list of primers and antibodies are provided as Supplementary Tables and included in the Supplementary Data file.

**Statistics and reproducibility**. Representative images shown as main figures were independently reproduced in at least two independent light stimulation/differentiation and confirmed in three individual wells for each experiment. Images included as supplementary information were reproduced in at least two independent stimulation/differentiation. Single-cell imaging quantification have been reduced with similar results in three independent images for at least two biological independent samples. qRT-PCR data have been reproduced at least in two biological independent samples. The scRNA-seq dataset is composed of two biological independent samples.

**Reporting summary**. Further information on research design is available in the Nature Research Reporting Summary linked to this article.

## Data availability

The scRNA-seq data generated in this study have been deposited in the GEO database under accession code GSE163505. The mouse brain developmental atlas used in this study from La manno et al., was downloaded from http://mousebrain.org/downloads.html. The human fetal hypothalamic dataset used in this study from Zhou et al. is available on GEO under the GSE118487. All other relevant data supporting the key findings of this study are available within the article and its Supplementary Information files or from the corresponding author upon reasonable request. Source data are provided with this paper.

## Materials availability

Plasmids are available on ADDGENE and cell lines upon request to the corresponding author.

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

## Acknowledgements

The authors would like to thank Zeeshan M. Ozair for discussion, criticisms, advise, and comments on the manuscript. Manon Valet for computational suggestions and Sandra Moreu for supporting plasmids generation. We also would like to thank Jean-Marx Santel and Peter Ingrassia for revising the manuscript. We are grateful to Brivanlou and Siggia lab members for inputs and criticism. We also would like to thank the Genomics Resource Center and the Bio-Imaging Resource Center for technical support. We wish to thank Kavli-foundation for supporting this research with the Kavli-NSI-Pilot grant to R.D.S. and A.H.B. R.D.S. was supported by EMBO-LTF-254-2019.

## Author contributions

R.D.S. conceived the project, performed experiments and analyzed the data. F.E. contributed to experimental discussions, imaging analysis and to establish the LED-light setup. E.A.R. contributed to scRNA-seq analysis. A.H.B. supervised the project and secured funding. R.D.S. and A.H.B. wrote the manuscript with input from all authors.

## Competing interests

A.H.B. is the co-founder of RUMI Scientific, RUMI Viro and OvaNova. A.H.B., F.E., and E.A.R. are shareholders of RUMI Scientific and RUMI Viro. R.D.S. has no competing interests to disclose.
