## [Peer Review File · Nature Communications]

Self-organization of human dorsal-ventral forebrain structures by light induced SHHReviewers' Comments:

Reviewer #1:

Remarks to the Author:

De Santis et al. applied a photoactivatable Cre-loxP DNA recombination system to understand embryogenesis. They demonstrated that the photoactivatable DNA recombination system can geometrically confine the expression of SHH in a monolayer culture of neuralizing hESCs, thereby inducing self-organizing centers of human hypothalamic structures in vitro. The reviewer evaluates that this work provides a powerful tool for experimental embryology and could facilitate our understanding of embryogenesis. Some minor comments are raised as follows:

Minor comments:

1. The authors are needed to add experimental details about how gene transduction of hESCs were performed using the piggyBac transposon system in the material and methods section.
2. Some typographical errors, such as XX, XY, Lox-P, are needed to be corrected.
3. The name "Opto-piggyBac" appears to be pretty confusing. This is because the tool presented in this study is mainly based on a photoactivatable Cre-loxP DNA recombination system or a photoactivatable Cre recombinase, previously named PA-Cre. The author did not directly modify the piggyBac transposon system itself. The piggyBac transposon system is just used for gene transduction of hESCs by the vectors for the photoactivatable Cre-loxP DNA recombination system. The reviewer suggests reconsideration of the name "Opto-piggyBac".

Reviewer #2:

Remarks to the Author:

The study by De Santis et al., is focused on improving the induction method of in vitro model system using transposon and optogenetic-based manipulation and showed a proof-of-principle experiment by inducing morphogen Shh to form what resembles the ventral forebrain (validated by traditional immunostaining and single-cell RNA-sequencing (scRNA-seq)). Although the methodological approach taken by the authors looks solid overall and could potentially improve the field, the current manuscript is not suitable for publication. The figures are unclear, methods are inadequately described, and insufficient controls are used for other researchers to use this method over any other methods that are currently available in an in vitro model system. Specific concerns are listed below.

1. The manuscript in its current form doesn't really describe a substantially novel method. Although integration of transposon and light-inducible Cre system (based on Magnet-Cre from Kawano et al., 2016 as mentioned by the authors), with an additional safety measurement through the use of a doxycycline-inducible promoter to induce gene expression is nice, I see very little improvement on existing techniques. Will this approach generate a different or more robust effect than standard methods (e.g. Shh-soaked beads) that can also generate artificial gradients? While the precise temporal control associated with optogenetic activation is a potential advantage, it is not clear whether this really contributes anything in this context. The authors might want to consider either validating the regulation and effects of Shh induction in more detail, or combine this with with another light-inducible split-Cre system (red/far-red system, i.e. Wu et al., 2020, Yen et al., 2020) and induce two-dimensional morphogen gradients (i.e. induction of dorsal and ventral morphogenic signals)?
2. Many controls are missing. First, there is no DOX-negative control with light exposure included, and it's thus hard to judge the efficiency of DOX promoter. Second, there is no quantification of induction efficiency in the light-exposed area stimulated (% of Neon-Green/Red). Third, there is no quantification of cells that express Nkx2-1 or Foxa2 but do not co-express light-induced Neon-Green fluorescence at DIV7 and DIV14. Other important details are also missing. What was the rationale of 24 hr blue-light stimulation (with repeated on/off cycle)? What is the minimum light-stimulation to induce Cre-lox recombination? What is the side-effect of prolonged light-stimulation? Did the authors

observe any change in proliferation or apoptosis (Zhuang et al., 2018, Yuan et al., 2017, Seko et al., 2011)?

3. The study does not contain enough novel findings to really qualify as a biology paper. Shh has been shown to induce ventral forebrain characteristics in numerous previous studies, but Shh induction in vitro induce both ventral diencephalic/tuberal hypothalamic (Foxd1/Rax/Nkx2-1) and telencephalic (Foxg1/Nkx2-1) identity. However, no telencephalic features are described in this study, and it is not clear exactly what is being generated here. The authors should take a closer look at the Nkx2-1-expressing cluster from the scRNA-seq dataset, which may include telencephalic cells.

4. The scRNA-seq analysis is inadequate. While comparing to previously published stage-matched datasets (La Manno, et al. 2020; Zhou et al., 2020) is appropriate, many details related to data analysis are missing, especially when the data is compared to the GW10 human scRNA-seq dataset. Other than a few transcription factors (Nkx2-1, Six3, Six6, Otp, and Rax), what are the other key genes that correlate to the published human dataset? Do the authors observe similar trajectory patterns with RNA velocity, the same pseudotime changes in gene dynamics, and what hypothalamic subregions are induced in this model? No supplementary dataset (of cluster-specific/enriched differential expressed genes) is provided, and the biological significance of the 32-odd clusters shown in Fig S3 is unclear.

Minor points:

1. The authors should consider depositing the plasmid in the publicly available repository (e.g. Addgene).
2. Heatmaps in Fig S2, x- and y-axis labels are unreadable. Fonts are too small.
3. In the Methods section entitled 'scRNA-Seq analysis', Nova-Seq (XX cycles), please indicate the number of cycles used.
4. Scanpy has already been published. Please cite the published manuscript.
5. Not enough detail is included as to how scRNA-Seq public datasets were used to compare to the in vitro dataset shown in the manuscript. For the correlation plot, were only highly-expressed genes used?
6. In Fig S2, please export image files from ABA, and add a label for gene names.
7. In Fig S2A - Nkx2.1 + FoxA2 expression domain is shown in magenta, but the MGE area should not express Foxa2.
8. In Fig S3, clusters are not shown. What do these clusters indicate in Fig 2. These clusters should be shown in the manuscript, and all DEGs should be shown in supplementary data. Key cluster markers are not very different, indicating that the number of clusters is too high, and not biologically meaningful.
9. No method is made as to how Fig S3B was generated. Functions are shown in the method, but detailed parameters are not shown. If default parameters were used, it should be stated in the Methods.
10. In Fig S3C, D Heatmaps are not readable.
11. Fig S3E legend is missing

Reviewer #3:

Remarks to the Author:

The manuscript entitled "Self-organization of human hypothalamic structures by light induced SHH" reports the response of iPSC-derived neural culture to a light-inducible SHH-expressing vector. The results are lacking controls and quantification needed to support the conclusion made by the authors. These are essential for the work to be rigorous enough for further consideration.

They first stably transfect an iPSC line with a Opto-piggyBAC light- and Dox-inducible CRE and a vector that expresses dsRED flanked by Lox sites and followed by NG (Neon Green – could do with clearer reference) and a T2A site allowing them to add another coding sequence after NG. The authors show that induction of NG expression in a specific area of the culture dish can be robustly achieved. They then made a second iPSC line containing the Opto-PiggyBAC and an expression vector containing

SHH following the T2A site. They show that induction of expression of NG in well-defined small stripe in the culture is also robustly achieved with this line. They then use this line to evaluate whether medio-lateral patterning of the neural culture can be achieved via these induced stripes of SHH-expressing cells. The 'induction' observed is defined by expression of a very small number of markers and can only be detected at day 14 of culture. There are a few major limitations to the results casting doubts on their interpretation. We list them below.

Major points

- Lack of a control culture. The study needed to be done comparing an induced stripe of NG-SHH with an induced stripe of NG alone (first cell line produced). Are control NG-induced cultures self-organising too and how? They need to get staining and scRNAseq of this essential control.
- Green is taken as proxy for Hh but no data shows Shh staining (or shh expression by any other method) needed to prove there is SHH expression and to measure how dynamic/where it is. Showing Hh expression and its correlation to NG expression in the induced culture is essential.
- What is the explanation for no Nkx2.1 lateral induction at day 7 (Figure 2C)? This is a substantial time after expression of the signal. In human embryos, Nkx2.1 does not take days to be expressed following expression of SHH in the ventral mesoderm. This casts a doubt on whether what we see is induction at all. Expression away from the green cells can be due to population expansion from initial proliferating Nkx2.1+ cells at the border of the induced stripe. Especially that in day 14, Pax6 is still detectable in cells next to boundary of the induced stripe. Cell culture is not a static system.
- Quantification is needed of the level of expression of markers per nuclei across the medio-lateral axis for all data presented (intensity of markers in intensely green, weak green and not green cells in the stripe and in the induced domain). For instance, without quantification, they can't prove that SHH(high) is associated with FOXA2 vs SHH(low) with NKX2-1.
- If the culture is, as expected, neural anterior, what is the reason for no MGE progenitor identity? This is very unexpected as dorsal forebrain identity is present and even quite a lot of neural crest (Figure 3B)?

Minor points

- Very intriguing cell organisation in the stripe of NG-SHH expressing cells. What are the clusters about (even at day 7) and does this happen in the NG only control too?
- Just because the signalling is in the same area it doesn't mean it's autocrine. From looking at merge images of Figure 2B most have either green or yellow so it's not in the same cell.
- The scale bars on the images in Supl Fig 1B seem wrong. Or else the widths of the induced areas are twice as large as the light permissive regions.
- Clarify when SMADs inhibition is added and removed, as well as when Dox is removed within figure 2 schematic
- Many grammatical errors and typos, even keeping XX and xxx in scRNAseq analysis method section.
- Figure 3 C and D are unreadable

Rebuttal Nature Communication, De Santis et al.

We would like to thank the reviewers for their constructive criticisms and suggestions which have subsequently helped us improve our paper. Taking the reviewers' recommendations under consideration, we have added 8 new panels as main figures, 5 new supplementary figures and two new supplementary tables. Here, we present a point-by-point response to each reviewer's specific points.

REVIEWER COMMENTS

Reviewer # 1:

We would like to thank the reviewer for her/his positive outlook on our work: *"... this work provides a powerful tool for experimental embryology and could facilitate our understanding of embryogenesis."*

The reviewer also has 3 minor comments, each addressed below:

1. *"The authors are needed to add experimental details about how gene transduction of hESCs were performed using the piggyBac transposon system in the material and methods section"*.

Following the reviewer's advice, we now provide further details of the experimental protocols, as well as the original references (Lacoste et al., 2009, Rosa et al., 2014) for our gene delivery system. These additional details can be found in the Material and Methods section.

2. *"Some typographical errors, such as XX, XY, Lox-P, are needed to be corrected."*

We have corrected any typographical errors.

3. *"... reconsideration of the name "Opto-piggyBac"*.

Following the reviewer's suggestion, we have now removed the name "Opto-piggyBac" from the text, and state that we are using a photoactivatable Cre-loxP DNA recombination system adapted from Kawano et al., 2016.

Reviewer # 2:

While the reviewer has a positive comment: *"...the methodological approach taken by the authors looks solid overall and could potentially improve the field,"*, she/he also express specific concerns, which we address point by point below:

1. *"The manuscript in its current form doesn't really describe a substantially novel method. Although integration of transposon and light-inducible Cre system (based on Magnet-Cre from Kawano et al., 2016 as mentioned by the authors), with an additional safety measurement through the use of a doxycycline-inducible promoter to induce gene expression is nice, I see very little improvement on existing techniques. Will this approach generate a different or more robust effect than standard methods (e.g. Shh-soaked beads) that can also generate artificial gradients?"*

Our experimental setup provides a simplified tool that pairs human embryonic stem cells with light-induced morphogens using a photoactivatable Cre-loxP DNA recombination system. To the best of our knowledge, activation of a light-induced morphogen that leads to self-organization of hESCs into ventral fates has never been reported before. Light-induced CRE-recombinases are mainly used upon transient transfection in cell lines and displays different background activity in the dark depending on the expression levels and transfection efficiency (Taslimi et al., 2016; Kawano et al., 2016; Meador et al., 2019; Morikawa et al., 2020). This represents an important limitation when assessing morphogen-induced cell fates, which are tightly linked to the level and distance from the source. Our approach overcomes these limitations, and provides standardization, scalability, tight regulation of expression and low background (Fig 1, Suppl. Fig 1). While we have tremendous respect for traditional experimental embryology tools, such as grafting beads, which we have used over decades, we also appreciate their limitations including little control over the extent of the inductive fields and secondary effect due to mechanical interferences and morphogenetic movements, especially for long-term measurements. Our light-induction of molecular signaling pathways in hESCs provides a more accurate spatial control, in line with more modern molecular embryology tools. To place the advantages of this method compared to experiment with beads, we now discuss this point in the introduction and the discussion.

2- “While the precise temporal control associated with optogenetic activation is a potential advantage, it is not clear whether this really contributes anything in this context. The authors might want to consider either validating the regulation and effects of *Shh* induction in more detail, or combine this with another light-inducible split-Cre system (red/far-red system, i.e. Wu et al., 2020, Yen et al., 2020) and induce two-dimensional morphogen gradients (i.e. induction of dorsal and ventral morphogenic signals)?”

Inducing two-dimensional gradients is an exciting possibility and an important application of our technology. However, we believe that is beyond the scope of this paper and we propose it as a future direction in the discussion.

3. “Many controls are missing. First, there is no DOX-negative control with light exposure included, and it’s thus hard to judge the efficiency of DOX promoter. Second, there is no quantification of induction efficiency in the light-exposed area stimulated (% of Neon-Green/Red). Third, there is no quantification of cells that express *Nkx2-1* or *a2* but do not co-express light-induced Neon-Green fluorescence at DIV7 and DIV14. Other important details are also missing. What was the rationale of 24 hr blue-light stimulation (with repeated on/off cycle)? What is the minimum light-stimulation to induce Cre-lox recombination? What is the side-effect of prolonged light-stimulation? Did the authors observe any change in proliferation or apoptosis (Zhuang et al., 2018, Yuan et al., 2017, Seko et al., 2011)?”

We thank the reviewer for prompting us to add more controls and quantifications, which is in line with reviewer #3 (see below). We addressed all 3 points. First, following the reviewer’s suggestions, we have now included in the new Fig 1B-C-D-E, new Suppl. Fig 1B-E, images and quantifications showing that DOX-negative control cells stimulated with light display less than 1% of NG induced cells (Fig 1-B-C-E). Second, we have now added the quantification of induction efficiency and show that the most efficient blue-light induction is accomplished in a 600 cycles of light illumination (24-hour), where 78.3 % of cells are NG positive, and this occurs without increase in cell death as quantified by CASP3 staining (Suppl. Fig 1E). Finally, we have now included single-cell fluorescence quantification for NKX2-1, FOXA2, and NG (exogenous SHH) at different time points during differentiation, day 2, 7, and 14 (Fig 2C-E). Single cell quantification demonstrates the non-cell-autonomous induction of ventral fates by light induction of SHH and displays fate acquisition dynamics from a light polarized SHH source in 2D (new Fig 2D-E-F).

4. The study does not contain enough novel findings to really qualify as a biology paper. *Shh* has been shown to induce ventral forebrain characteristics in numerous previous studies, but *Shh* induction in vitro induce both ventral diencephalic/tuberal hypothalamic (*Foxd1/Rax/Nkx2-1*) and telencephalic (*Foxg1/Nkx2-1*) identity. However, no telencephalic features are described in this study, and it is not clear exactly what is being generated here. The authors should take a closer look at the *Nkx2-1*-expressing cluster from the scRNA-seq dataset, which may include telencephalic cells.

We have included in the revised manuscript an additional scRNA-seq replica (new Fig 3B, new Suppl. Fig 5A-B). Using the integrated dataset of biological replicates, we identified four distinct cellular classes among NKX2-1+ cells: hypothalamic, telencephalic, proliferating ventral progenitors and OTP hypothalamic neurons that express lineage-specific marker genes (Fig 3E-F-G, Suppl. Table 1). We further validated by immunostaining FOXG1 (telencephalic) and SIX6, RAX and OTP (hypothalamic) markers revealing that the telencephalic and hypothalamic populations are distributed along with proximal-distal coordinates from the SHH source (Fig 3F-G-H). We showed that the geometrical confinement of SHH by light stimulation induces both telencephalic and hypothalamic cells that are spatially segregated from the morphogenic signaling source, which is supported by scRNA-seq and immunostaining analysis.

5. “The scRNA-seq analysis is inadequate. While comparing to previously published stage-matched datasets (La Manno, et al. 2020; Zhou et al., 2020) is appropriate, many details related to data analysis are missing, especially when the data is compared to the GW10 human scRNA-seq dataset. Other than a few transcription factors (*Nkx2-1*, *Six3*, *Six6*, *Otp*, and *Rax*), what are the other key genes that correlate to the published human dataset? Do the authors observe similar trajectory patterns with RNA velocity, the same pseudo-time changes in gene dynamics, and what hypothalamic subregions are induced in this model? No supplementary dataset (of cluster-specific/enriched differential expressed genes) is provided, and the biological significance of the 32-odd clusters shown in Fig S3 is unclear.”

We present a more comprehensive cross-comparison of our *in vitro* dataset and the human fetal reference from Zhou et al., (2020). We computed the active gene regulatory networks (regulons) in each dataset using pySCENIC (Van de Sande et al., 2020). Among the 427 active regulons identified in the human fetal hypothalamus dataset, the 72.5% (310 regulons) are shared with our *in-vitro* dataset (Suppl. Fig 8B, Suppl. Table 2). Gene-set analysis of the 310 shared regulons using the “Tissues protein expression from the human proteome map”, display

the significant enrichment of the “fetal brain” category (adjusted p-val 0.002). This analysis shows a significant overlap of gene regulatory network between our *in vitro* dataset and human brain development. Additionally, to match *in vitro* gene expression patterns with the recently described human hypothalamic fetal dataset from Zhou et al., (2020), we computed pseudotime trajectories using RNA velocity. We identified a trajectory from ventral proliferating cells to ventral OTP hypothalamic neurons (new Fig 3B). Gene-set enrichment analysis reveals three different modules related to cell cycle, progenitor maintenance and neuronal maturation along this trajectory (Suppl. Fig 8B). We explored whether some important gene expression patterns recently described in the context of human fetal hypothalamic development were recapitulated in our model. We revealed the dynamics of TTYH1, HMGA2 and MYBL2, which recapitulate the same trends observed in the human fetal hypothalamus at PCW 10 by Zhou et al., 2020 (Suppl. Fig 8C). These findings strengthen confidence in our *in vitro* model which we believe faithfully recapitulates gene expression patterns that occur during human fetal hypothalamic development. Finally, Light-induced SHH during neural induction generates distinct ventral neural progenitor populations, representative of the mammillary hypothalamus NKX2-1+/SIX6+/SIX3+/RAX+ (Fig 3B-C-E, Suppl. Fig 5C, Suppl. Table 1) (Shimogori et al., 2010; Martinez-Ferre and Martinez, 2012; Morales-Delgado et al., 2014), a more anterior telencephalic population positive for NKX2-1+/FOXP2+ (Fig 3E-F-H) (Maroof et al., 2013; Cederquisted et al., 2019; Rifes et al., 2020), and a small population of ventral neurons positive for NKX2-1, OTP, NHLH2 representative of the peri-mammillary hypothalamic neurons (Martinez-Ferre and Martinez, 2012; Morales-Delgado et al., 2014). Importantly, these populations are spatially distinct and self-organized along a proximal-distal axis (Fig 3F-G-H). As suggested by the reviewer, we included in the revised manuscript a supplementary table with the differentially expressed genes enriched in each cluster (Suppl. Table 1). We revised the heatmaps to better visualize in the manuscript to show the integration of our in-vitro dataset with an in-vivo mouse brain atlas from La Manno et al., 2020. (Suppl. Fig 6B-C).

Minor points:

1. “The authors should consider depositing the plasmid in the publicly available repository (e.g. Addgene).”
We deposited all our plasmid collection in ADDGENE.

2. “Heatmaps in Fig S2, x- and y-axis labels are unreadable. Fonts are too small.”
We updated all heatmaps in the manuscript (Suppl. Fig 6B-C).

3. “In the Methods section entitled ‘scRNA-Seq analysis’, Nova-Seq (XX cycles), please indicate the number of cycles used.”
We updated that Material and Methods section with the correct number of cycles (NovaSeq 6000 SP flowcell as 28 x 94 x 8).

4. “Scanpy has already been published. Please cite the published manuscript.”
We included the appropriate reference (Wolf, F. A., Angerer, P. & Theis, F. J. SCANPY: large-scale single-cell gene expression data analysis. *Genome Biol* 19, 15 (2018).

5. “Not enough detail is included as to how scRNA-Seq public datasets were used to compare to the in vitro dataset shown in the manuscript. For the correlation plot, were only highly-expressed genes used?”
The correlation analysis was done using the high variable genes only. We included more details in the Material and Methods section.

6. “In Fig S2, please export image files from ABA, and add a label for gene names.”
We re-exported the original images from the Allan Brain Atlas (ABA) as suggested by the reviewer (Suppl. Fig 2A).

7. “In Fig S2A - Nkx2.1 + FoxA2 expression domain is shown in magenta, but the MGE area should not express Foxa2.”
The new figure 3E-F-G-H has now clarified this point by showing scRNA-seq analysis and immunostaining of telencephalic and hypothalamic markers.

8. “In Fig S3, clusters are not shown. What do these clusters indicate in Fig 2. These clusters should be shown in the manuscript, and all DEGs should be shown in supplementary data. Key cluster markers are not very different, indicating that the number of clusters is too high, and not biologically meaningful.”
We have now included a Supplementary Table 1 reporting the differentially expressed genes and replot marker genes for each cluster in the dataset. (Suppl. Fig 5C, Supplementary Table 1)

9. “No method is made as to how Fig S3B was generated. Functions are shown in the method, but detailed parameters are not shown. If default parameters were used, it should be stated in the Methods.”

We performed dataset integration using “scanpy.tl.ingest” using default parameters and the correlation person analysis also using default parameters, that are stated in the new material and method section (“Analysis and integration of publicly available scRNA-seq datasets”).

10. “In Fig S3C, D Heatmaps are not readable.”

We have updated all the heatmaps in the manuscript to make them more readable.

11. “Fig S3E legend is missing”

We updated the figure legend descriptions according to the revised manuscript.

Reviewer # 3:

This reviewer has several concerns that we address individually below:

Major points

1- “Lack of control culture. The study needed to be done comparing an induced stripe of NG-SHH with an induced stripe of NG alone (first cell line produced). Are control NG-induced cultures self-organising too and how? They need to get staining and scRNAseq of this essential control.”

Following the reviewer’s suggestion, we have now performed side-by-side experiments comparing light induced Neon-Green-T2A-SHH (NG-T2A-SHH) to the control (NG-CNTRL) at different time points during differentiation (d2, d7, d14). The NG-CNTRL control does not induce ventral marker genes or SHH targets genes as shown by immunostaining and RNA analysis. These results have now been added to new Figure panels (Fig 2B-C, Fig 3D-H, Suppl. Fig 2B, Suppl. Fig 3B-C), and demonstrate that induction and self-organization of ventral neural fates is specific to SHH activation.

2- “Green is taken as proxy for Hh but no data shows Shh staining (or shh expression by any other method) needed to prove there is SHH expression and to measure how dynamic/where it is. Showing Hh expression and its correlation to NG expression in the induced culture is essential.”

The Green expression module is based on the production of a bicistronic mRNA that encodes Neon-Green and SHH separated by a T2A sequence (Wang et al., 2015; Liu et al., 2017)(Fig 2A). Thus, upon light induction, the bicistronic mRNA generates two independent proteins. This experimental setup guarantees the co-expression of both proteins in the same cell, as documented in many studies over the years (Wang et al., 2015; Liu et al., 2017). Nevertheless, to independently address the reviewer’s concern, we now show that reducing the area of illumination reduces the expression of Neon-Green, SHH, and its downstream target GLI1 as measured by qRT-PCR. This has now been added to a new figure panel (Suppl.Fig 4C).

3- “What is the explanation for no Nkx2.1 lateral induction at day 7 (Figure 2C)? This is a substantial time after expression of the signal. In human embryos, Nkx2.1 does not take days to be expressed following expression of SHH in the ventral mesoderm. This casts a doubt on whether what we see is induction at all. Expression away from the green cells can be due to population expansion from initial proliferating Nkx2.1+ cells at the border of the induced stripe. Especially that in day 14, Pax6 is still detectable in cells next to boundary of the induced stripe. Cell culture is not a static system.”

There seems to be a slight misunderstanding from the reviewer that we would like to respectfully address. First, Nkx2.1 is detected starting at day 7, with a small lateral domain that expands over time and it is maintained at day 14 (Fig 2D-E-F, new Suppl.Fig 4A). We now provide additional evidence of ventral *neural* fate induction upon light modulation of SHH. We show that the ventral marker FOXA2 is induced as early as day 2 in both a cell-autonomous and non-cell-autonomous manner (new Fig 2B-C, Suppl. Fig 2B-C). Ventral neural fates self-organize to generate a population of cells expressing different combinations of ventral markers (FOXA2 and NKX2-1) at days 7-14 that over time acquire spatial coordinates (new Fig 2B-C-D-E-F, new Suppl. Fig 3A, new Suppl. Fig 4A). In conclusion, our experiments describe the self-organization of ventral structures over 14 days, where ventral fates are induced as early as day 2. We speculate that direct signaling in combination with proliferation/expansion of progenitor pool is the driver of self-organization in our *in vitro* structure. We have included this specific point raised by the reviewer in the new discussion.

4- “Quantification is needed of the level of expression of markers per nuclei across the medio-lateral axis for all data presented (intensity of makers in intensely green, weak green and not green cells in the stripe and in

the induced domain). For instance, without quantification, they can't prove that SHH(high) is associated with FOXA2 vs SHH(low) with NKX2-1.

Following on the reviewer's suggestion we have now quantified the single-cell intensity profiles of NKX2-1, FOXA2, and NG (reporter gene of exogenous SHH) by immunostaining. We have also amended our previous claim on the level of exogenous SHH (NG) and its correlation with the FOXA2/NKX2-1 specification, since single-cell quantification shows no correlation (Fig 2C-D-E). In conclusion, single-cell imaging quantification described the non-cell-autonomous induction of both FOXA2⁺ and NKX2-1⁺ cells over time upon light-induced SHH.

5- "If the culture is, as expected, neural anterior, what is the reason for no MGE progenitor identity? This is very unexpected as dorsal forebrain identity is present and even quite a lot of neural crest (Figure 3B)?"

We re-analyzed our scRNA-seq dataset looking for MGE progenitors (NKX2-1⁺/FOXG1⁺) of the anterior ventral telencephalon. scRNA-seq and immunostaining revealed the presence of cell positive for NKX2-1⁺/FOXG1⁺, MGE progenitors, that are juxtaposed to the light induced organizer and distinct from the hypothalamic (NKX2-1⁺SIX6⁺) population (new panel Fig 3E-F-G-H).

Dorsal forebrain populations are induced by dual-SMAD inhibition in the most distal position from the SHH organizer. PAX6 negative non-neural ectoderm cells, such as neural crest and epidermis, mainly arises from well edge far away from the SHH organizer (new Suppl. Fig 4B).

Minor points

1- "Very intriguing cell organisation in the stripe of NG-SHH expressing cells. What are the clusters about (even at day 7) and does this happen in the NG only control too?"

We agree that cells clusters are intriguing, and we speculate that is probably due to cell-cell interaction of the different cell types generated by the SHH organizer (floor plate, hypothalamic and telencephalic progenitors). This peculiar cell organization is specifically induced by the NG-T2A-SHH but not in the control NG-CNTRL (new Fig 2D, new Suppl. Fig 3B).

2- "Just because the signalling is in the same area it doesn't mean it's autocrine. From looking at merge images of Figure 2B most have either green or yellow so it's not in the same cell."

We agree with the reviewer and have now included single cell quantification and high-magnification immunostaining with single cell resolution to show that at day-2, FOXA2 positive cells are induced in both autocrine and paracrine manner (Fig 2B-C, Suppl. Fig 2C).

3- "The scale bars on the images in Suppl. Fig 1B seem wrong. Or else the widths of the induced areas are twice as large as the light permissive regions."

We corrected the scale bar and included more images from a 96 well plate culture experiment (Suppl. Fig 1B).

4- "Clarify when SMADs inhibition is added and removed, as well as when Dox is removed within figure 2 schematic"

We clarified this point in the main text, in the figure legend and in the Material and Methods.

5- "Many grammatical errors and typos, even keeping XX and xxx in scRNAseq analysis method section."

We apologize for our typos and shortcomings regarding grammar, which we have now corrected. We have also added clarification on XX and XY (female and male) lines used in our study. Finally, we have also updated the GEO dataset number (GSE163505) in the material and methods of the manuscript.

6- "Figure 3 C and D are unreadable"

We replotted the heatmaps in the revised manuscript to make them easier to read (Suppl. Fig6 B-C).

Rebuttal references that are not present in the main text:

-Liu, Z. et al. Systematic comparison of 2A peptides for cloning multi-genes in a polycistronic vector. *Sci Rep-uk* 7, 2193 (2017).

-Meador, K. et al. Achieving tight control of a photoactivatable Cre recombinase gene switch: new design strategies and functional characterization in mammalian cells and rodent. *Nucleic Acids Res* 47, e97–e97 (2019).

-Taslimi, A. et al. Optimized second-generation CRY2–CIB dimerizers and photoactivatable Cre recombinase. *Nat Chem Biol* 12, 425–430 (2016).

-Wang, Y., Wang, F., Wang, R., Zhao, P. & Xia, Q. 2A self-cleaving peptide-based multi-gene expression system in the silkworm *Bombyx mori*. *Sci Rep-uk* 5, 16273 (2015).

We thank all 3 reviewers for their constructive criticism, which we believe has strongly improved the quality of our manuscript. In addition, the critiques have also provided us with a unique opportunity to study models of human embryonic neural development and patterning events, during stages of development that would otherwise be impossible to scrutinize. We sincerely hope that they will now find it to be acceptable for publication.

Reviewers' Comments:

Reviewer #1:

Remarks to the Author:

I confirm that the authors have responded to all the concerns that I raised in the first round of reviewing.

Reviewer #2:

Remarks to the Author:

The revised manuscript is much improved, and has addressed almost all my outstanding concerns. A few discrepancies in the classification of hypothalamic progenitor subtypes needs to be corrected, however. The authors refer to Nkx2.1/Rax/Six3+ cells throughout as "mammillary", but this is not correct. These are tuberal hypothalamic progenitors, as described in Shimogori, et al. 2010 and more recently using scRNA-Seq from embryonic mouse hypothalamus by Kim, et al. 2020. The use of "perimammillary" to refer to Nkx2.1/Nhlh2/Otp+ cells is likewise confusing, as these same studies suggest that these are premammillary neural precursors. Aside from this, however, I have no further concerns.

Reviewer #3:

Remarks to the Author:

The revised manuscript is much improved, with most concerns have been satisfactorily addressed. However, a couple of main concerns are still unresolved:

- I am still very puzzled by the very weak number of cells nkx2.1+ at day 7. Together with the fact that GFP is lost over time in quite a few of the induced cells, I am worried that the results at day 14 is created by proliferation dynamics rather than by signalling. This main concern still needs to be addressed properly by time-lapse imaging . It is important to prove that what is seen at day 14 results from induction from a signalling population to a receiving field.

- The identity of the forebrain tissue obtained is not clearly defined. The Day 14 scRNAseq shows (Fig. 4B) that it is mostly diencephalic (hypothalamus/thalamus) with a very small cluster labelled 'ventral telencephalic progenitors'. However, Fig4F shows a complete field of cells positive for Foxg1 and Nkx2.1 at day14. Foxg1 expression is specific to telencephalon and nasal retina. The authors show that cells are negative for RAX and Six6 so they are not retinal, suggesting they are in vast majority ventral telencephalic. This strongly contradicts the scRNAseq dataset. It may suggest difficulties for cells to 'decide' of their AP identity with possible mixed identity. The authors need to stain the culture for Foxg1, hypothalamic and thalamic markers and measure whether nuclei are double positive for distinct AP markers. The key question is whether there is truly organisation or weird cells with mixed/confused identities. ThescRNAseq would not resolve this (as analysis take two-five most abundantly expressed genes to establish 'identities') so staining is required and picture of close-up and whole field around the GFP+ cell stripe.

- Related to this second point, there is only one picture showing a broader field (Suppl. Fig 7B) and this shows very patchy expression pattern of Pax6. Proper quantification of spatial organisation of cell identity is needed.

Second rebuttal Nature Communication, De Santis et al.

We would like to thank the reviewers for their constructive criticisms and suggestions.

We are pleased that reviewers #1 and #2 find our revised manuscript “much improved” and that their concerns have been addressed via minor adjustments. Following the positive comments of reviewer #3, we have now performed additional experiments, included in the revised manuscript as supplementary figures, addressing the dynamics of induction of our light-inducible setup by time-lapse imaging and specific marker genes expression by single nuclei imaging quantification.

Here, we present a point-by-point response to each reviewer’s specific points.

Reviewer #1 :

This reviewer#1 is satisfied with our revisions and we are grateful to her/his comment: ***“I confirm that the authors have responded to all the concerns that I raised in the first round of reviewing”.***

Reviewer #2 :

We are also grateful to reviewer#2 who states: ***“The revised manuscript is much improved, and has addressed almost all my outstanding concerns.”***

The reviewer however, would also like a few minor corrections: ***“A few discrepancies in the classification of hypothalamic progenitor subtypes needs to be corrected, however. The authors refer to Nkx2.1/Rax/Six3+ cells throughout as “mammillary”, but this is not correct. These are tuberal hypothalamic progenitors, as described in Shimogori, et al. 2010 and more recently using scRNA-Seq from embryonic mouse hypothalamus by Kim, et al. 2020. The use of “peri-mammillary” to refer to Nkx2.1/Nhlh2/Otp+ cells is likewise confusing, as these same studies suggest that these are premammillary neural precursors.”***

In agreement with the reviewer point, and taking into consideration her/his comment, we re-evaluated the scRNA-seq classification and adjusted the labels in Ventral tuberal hypothalamic progenitors (NKX2.1+/RAX+/SIX6+/SIX3+) and Ventral hypothalamic neurons (NKX2-1+/NHLH2+/OTP+) in the revised manuscript (Fig. 3B, Suppl. Fig 5B-C).

The reviewer ends with: ***“Aside from this, however, I have no further concerns”.***

Reviewer #3:

We thank the reviewer#3 for her/his positive outlook and the statement: ***“The revised manuscript is much improved, with most concerns have been satisfactorily addressed”.***

The reviewer also expresses: ***“However, a couple of main concerns are still unresolved”.***

We address these below:

1- ***“I am still very puzzled by the very weak number of cells nkx2.1+ at day 7. Together with the fact that GFP is lost over time in quite a few of the induced cells, I am worried that the results at day 14 is created by proliferation dynamics rather than by signalling. This main concern still needs to be addressed properly by time-lapse imaging . It is important to prove that what is seen at day 14 results from induction from a signalling population to a receiving field”.***

In agreement with, and following the reviewer's suggestion, we have now performed the requested time-lapse imaging over the 14 days. This new data shows that the light stimulated domain is stably maintained over 14 days of differentiation, and presented in the **new Suppl. Fig 3D** and quantified in **Suppl. Fig 3E**.

2- “The identity of the forebrain tissue obtained is not clearly defined. The Day 14 scRNAseq shows (Fig. 4B) that it is mostly diencephalic (hypothalamus/thalamus) with a very small cluster labelled ‘ventral telencephalic progenitors’. However, Fig4F shows a complete field of cells positive for Foxg1 and Nkx2.1 at day14. Foxg1 expression is specific to telencephalon and nasal retina. The authors show that cells are negative for RAX and Six6 so they are not retinal, suggesting they are in vast majority ventral telencephalic. This strongly contradicts the scRNAseq dataset. It may suggest difficulties for cells to ‘decide’ of their AP identity with possible mixed identity. The authors need to stain the culture for Foxg1, hypothalamic and thalamic markers and measure whether nuclei are double positive for distinct AP markers. The key question is whether there is truly organisation or weird cells with mixed/confused identities. ThescRNAseq would not resolve this (as analysis take two-five most abundantly expressed genes to establish ‘identities’) so staining is required and picture of close-up and whole field around the GFP+ cell stripe”.

Following the advice of the reviewer, we co-stained our light-induced D-V structures at day 14 with telencephalic (FOXG1) and hypothalamic (NKX2-2) markers. This demonstrated that FOXG1 (telencephalic) domain is established outside of the SHH induced cells, that instead express NKX2-2 (hypothalamic). This is now presented in **the new Suppl. Fig7D**. As suggested by the reviewer, we also performed single nuclei quantification of FOXG1/NKX2-2 immunostaining to demonstrate distinct and specific expression of either FOXG1 or NKX2-2 (**new Suppl. Fig7E**). Taken together these new results demonstrate that the vast majority of the cells are un-ambiguously telencephalic or hypothalamic.

3- “Related to this second point, there is only one picture showing a broader field (Suppl. Fig 7B) and this shows very patchy expression pattern of Pax6. Proper quantification of spatial organisation of cell identity is needed”.

To address the reviewer's concern, we adjusted the contrast of the image to clearly visualize the homogenous expression of PAX6 and included its quantification with spatial coordinates from the edge of the well (**new Suppl. Fig 7B**).

We thank all 3 reviewers for their constructive criticism, which we believe has improved the quality of our manuscript. We sincerely hope that they will find this second round of revisions to be acceptable for publication.

Reviewers' Comments:

Reviewer #3:

Remarks to the Author:

I am now fully satisfied by the revised manuscript. Thanks to the authors for carefully considering my concerns and do the experiments needed to answer these.